# Reversible domain closure modulates GlnBP ligand binding affinity

**Qun Chen**[1☯], **Fang Li**[1☯], **Xiaobing Zuo**[2], **Jin Chen**[3], **Peiwu Qin**[1], **Chuhui Wang**[1], **Jin Xu**[4], **Danyu Yang**[4], **Baogang Xing**[5], **Ying Liu**[6], **Peng Jia**[6], **Linling Li**[7], **Chengming Yang**[5]*, **Dongmei Yu**[8]*

**1** Center of Precision Medicine and Healthcare, Tsinghua-Berkeley Shenzhen Institute, Shenzhen, Guangdong Province, China, **2** X-ray Science Division, Argonne National Laboratory, Argonne, IL, United States of America, **3** Institute of Chinese medical sciences, University of Macau, Macau, China, **4** Shenzhen Aier Eye Hospital, Futian District, Guangdong Province, China, **5** University of Science and Technology Hospital, Shenzhen, Guangdong Province, China, **6** Food Inspection & Quarantine Center, Shenzhen Custom, Shenzhen, Guangdong, China, **7** Shenzhen Maternity and Child Healthcare Hospital, Affiliated to Southern Medical University, Shenzhen, Guangdong, China, **8** School of Mechanical, Electrical & Information Engineering, Shandong University, Weihai, Shandong Province, China

☯ These authors contributed equally to this work.
* yudongmei@sdu.edu.cn (DY); hanks281128@163.com (CY)

**Data Availability Statement:** The data are available through the BMRB Data Bank, ID number: 50310.

**Funding:** Prof. Peiwu Qin received funding from the National Natural Science Foundation of China (grant numbers 31970742, 81972854, 81802626,

## Abstract

Glutamine binding protein (GlnBP) is an *Escherichia Coli* periplasmic binding protein, which binds and carries glutamine to the inner membrane ATP-binding cassette (ABC) transporter. GlnBP binds the ligand with affinity around 0.1μM measured by isothermal titration calorimetry (ITC) and ligand binding stabilizes protein structure shown by its increase in thermodynamic stability. However, the molecular determinant of GlnBP ligand binding is not known. Electrostatic and hydrophobic interaction between GlnBP and glutamine are critical factors. We propose that the freedome of closure movement is also vital for ligand binding. In order to approve this hypothesis, we generate a series of mutants with different linker length that has different magnitude of domain closure. Mutants show different ligand binding affinity, which indicates that the propensity of domain closure determines the ligand binding affinity. Ligand binding triggers gradual ensemble conformational change. Structural changes upon ligand binding are monitored by combination of small angle x-ray scattering (SAXS) and NMR spectroscopy. Detailed structure characterization of GlnBP contributes to a better understanding of ligand binding and provides the structural basis for biosensor design.

## Introduction

In Gram-negative bacteria, periplasmic ATP-binding cassette (ABC) transporter systems are responsible for transporting a broad variety of nutrients across the cytoplasmic membrane. ABC transporter consists of three components: a periplasmic binding protein, which shuttles the ligands through the outer membranes into the periplasmic spaces; an integral membrane protein complex that provides the transmembrane pathway; and two cytoplasmic nucleotide-

http://www.nsfc.gov.cn/english/site_1/index.html),
the Science, Technology and Innovation
Commission of Shenzhen Municipality (grant
number JSGG20191129110812, http://stic.sz.gov.
cn/), and the Southern University of Science and
Technology Hospital Dean Research Fund (2020-
A4). The funders had no role in study design, data
collection and analysis, decision to publish, or
preparation of the manuscript.

**Competing interests:** The authors have declared
that no competing interests exist.

binding domains that provide the driving power by hydrolyzing ATP molecules. The interaction between the protein transporter and ligand activates the energy-coupling transportation pathway and results in the opening of a channel and the subsequent translocation of the ligand across the cytoplasmic membrane [1, 2].

The first periplasmic binding protein was discovered by Pardee in 1996 [3]. Since then, many periplasmic binding proteins have been characterized and their structures have been studied extensively by x-ray crystallography and solution NMR [4–7]. The periplasmic binding proteins of ABC transporters from Gram-negative bacteria possess a common architecture (Fig 1). Glutamine binding protein (GlnBP), ATP-binding cassette (ABC) transporter, in this study is Escherichia Coli periplasmic binding protein of L-glutamine, which binds and carries glutamine to the inner membrane. The crystal structures of ligand-free and bound GlnBP show obvious conformational change upon ligand binding by structural alignment [4, 7]. GlnBP is classified in the same Structural Classification of Proteins SCOP family (phosphate-binding protein-like) as human GluR2, but shares 17% sequence identity with GluR2. GlnBP is comprised of two domains linked by rigid β strands, whose movement induces open-to-close conformation upon ligand binding. GlnBP undergoes conformational change due to the hinge bending and twisting motion [8]. The deep central cleft between domains provides the ligand-binding site with sub-micromolar affinity. The conformational changes upon ligand binding have been studied by the combination of chemical shift perturbation and 15N-1H scalar coupling [9]. The aliphatic side chain of the glutamine is sandwiched in a hydrophobic pocket formed between Phe13 and Phe50, which has 21 van der Waals contacts with GlnBP Lys115 and His156. These interactions are unique to GlnBP among known amino acid binding proteins, which apparently contribute to the ligand binding specificity of GlnBP. The hydrogen-bond networks increase the stability of GlnBP-ligand complex [4].

Ligand-bound GlnBP triggers the conformational change of downstream membrane transporter, but not ligand free GlnBP. Ligand binding modifies GlnBP's binding property with inner membrane transporter. GlnBP structural information is required to delineate GlnBP binding specificity and dynamic property. The use of GlnBP as glutamine biosensor has been proposed. Ligand is removed completely from GlnBP by GdnHCl and EGTA; then GlnBP is refolded on desalting column to remove GdnHCl. Ligand binding affinity is measured by isothermal titration calorimetry. However, the detailed knowledge of protein dynamics, binding

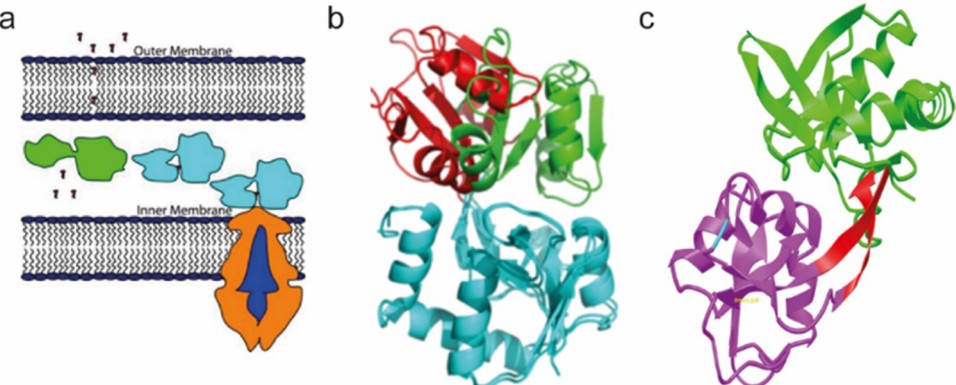

**Fig 1. Model of glutamine transportation and structure of GlnBP.** (a) Ligand-bound GlnBP interacts with membrane ABC transporter and induces the transportation of glutamine into the cytoplasm. (b) GlnBP has two domains and the interdomain rotation is demonstrated by the transition from green to red structure model while maintaining the position of another domain when glutamine binds to GlnBP. (c) The linker region of GlnBP is highlighted in red color.

affinity, and thermal stability is required for the development of sensor [8]. Here, we present the first small angle x-ray scattering study on GlnBP to monitor the gradual ensemble structural changes upon ligand binding. To map out determining factors for ligand binding, we generate a few mutants with different freedom and magnitude of domain closure. Our hypothesis is that domain closure is critical for ligand binding and this restriction of domain closure can be reversible controlled in a chemical manner. Mutants are created with site-directed mutagenesis to change the ligand affinity by introducing a linker on the opposite site of ligand binding pocket. The molecular weights of mutants are verified by mass spectrometry. We utilized different biophysical techniques to characterize the GlnBP wildtype and mutants. Thermal stability is studied with urea CD titration and Gibbs free energy is extracted from the CD urea titration curve. GlnBP conformational changes have been studied by NMR [5, 9], but the complete backbone assignment is still missing. Sedimentation velocity experiments of analytical ultracentrifuge conclude that GlnBP is monomeric and interference data shows the presence of aggregates ranging from ~0.1% at 83 μM to ~0.2% at 1480 μM. The data presented in this article is a further step towards understanding of the ligand binding, structural changes, and thermodynamics of GlnBP.

## Experimental methods

### GlnBP protein expression, purification, and removal of ligand

GlnBP gene is cloned into pET11a vector and the sequence is verified by DNA sequencing. Plasmid with GlnBP gene is transformed into BL21 star (DE3) competent cells for protein expression. Bacteria are cultured overnight in a 5 ml LB medium containing 100 μg/ml of ampicilin at 37˚C, and then cell pellet (minimize the volume of LB medium) is transferred into 1 L M9 minimal medium (6.78 g Na2HPO4, 3.0 g KH2PO4, 1 g NaCl, 2 g Glucose, 1 g 15N NHCl4, 2 mM MgSO4, 0.2 mM CaCl2, 1x MEM vitamin mix, 1x trace element solution, 100 μg ampicilin per liter medium) at 37˚C until OD600 reaches around 0.8. 1 mM IPTG is used to induce protein express for 3 hours at 37˚C. Cells are harvested at 5000 g for 30 min at 4˚C. The cell pellets are resuspended in the buffer (10 mM sodium acetate, pH 5.5) and broken by microfluidizer at 1200 psi. Clear supernatant is collected by centrifuge at 12,000 rpm for 30 min. Protein is purified by cation exchange (CM column) and GlnBP collections are pooled and buffer exchanged for gel filtration with 10 mM Tris and 300 mM NaCl. The fractions with GlnBP from gel filtration are pooled and exchanged to 10 mM Tris buffer with Amicon Ultra-10K (10,000 Da MWCO) device. Concentrated GlnBP run through cation exchange chromatography and elute with a linear gradient of 0–500 mM NaCl. The identity and integrity of the final protein is confirmed by SDS-PAGE. The ligand glutamine is dissociated from protein by treatment with 6 M guanidine hydrochloride at room temperature for 2 hours. Desalting column is equilibrated with 2 M and 6 M GdnHCl and GlnBP is run through the column to remove glutamine. The desalting column is thoroughly washed and equilibrated with 10 mM Tris. Guanidine is removed by running the desalting column with 10 mM Tris. 10 mM Tris (pH 6.8) buffer will be used for further NMR and ITC experiments, but 10 mM phosphate buffer will be used for CD urea titration because of the strong absorbance of Tris buffer.

### Isothermal titration calorimetry

GlnBP is dialyzed into buffer A (10 mM Tris at pH 7.2). Glutamine is dissolved into buffer A. Background titration is performed to check the dilution effect of glutamine. For optimal measurement of ITC, the product of protein concentration and association constant should stay in the range between 1 to 100. The protein concentration is 10–30 μM and ligand is 10 times concentrated about 300 μM. Around 1.8 ml protein is loaded into the sample cell with long syringe

and 292 μl ligand is loaded into injection needle with stirring to remove bubbles. First step injects 2 μl ligand into protein solution, and then 10 μl ligand is injected each time with 4 min delay between injections. The plot of DP (different power) with respect to the time is drawn by MicroCal software and thermal parameter and binding affinity are extracted by fitting the data.

## Circular dichroism

Urea denaturation is monitored by circular dichroism (CD) with an AVIV Model 62DS spectrometer. Experiments are conducted in 10 mm cuvettes with constant 2.0 mL volume. About 5–6 μM protein in 10 mM phosphate buffer (pH 7.5) is used in all the assays. Wavelength from 190 nm to 230 nm is scanned to find appropriate wavelength to monitor the protein structural changes. The dynode is kept below 500. The urea titration is from 0 M to 8 M with 0.1 M increment for 80 shots and 30 s averaging time. The sample is stirred at 78 rpm during the process of titration. The data is analyzed by Origin. The ellipticity data, corrected for dilution, are fit to Eq (1):

$$y = \frac{y_n + m_n[urea] + (y_u + m_u[urea]) \times \exp\left(-\left(\frac{\Delta G_0 - m[urea]}{RT}\right)\right)}{1 + \exp\left(-\left(\frac{\Delta G_0 - m[urea]}{RT}\right)\right)} \tag{1}$$

where $m_n$ and $y_n$ are the slope and intercept, respective, for the pre-transition baseline; $y_u$ and $m_u$ are the slope and intercept for the post-transitional baseline; $\Delta G_0$ is the Gibbs energy change for unfolding in the absence of urea; m describes the sensitivity of conformational free energy to urea concentration [10].

## Small angle X-ray scattering

SAXS is performed at Argonne National Laboratory sector 12-ID-B. Data are collected at room temperature on a 1024 x 1024-pixel CCD detector with sample-to-detector distance of 1 m and the transmission intensity is measured with a PIN diode beamstop. Protein solution scattering is conducted with concentration of 6.5 mg/ml in 10 mM Tris, pH 7.5. Background scattering from buffer is subtracted from all samples. The zero angle scattering intensity I(0) and overall radius of gyration Rg are obtained from a Guinier approximation to the low-q region of the scattering profiles satisfying the condition, Qmax* Rg < 1.3. The P(r) functions have the characteristic bell shape of globular complexes with well-defined maximum diameter size for all the samples prepared. I(q) is the scattered X-ray intensity per unit solid angle and q is the amplitude of the scattering vector, given by $4\pi(\sin\theta)/\lambda$, where $2\theta$ is the scattering angle and $\lambda$ is the wavelength of the scattered X-rays (0.98 Å). ΔP(r) manifests statistical significant variation in the distribution of the frequency of vectors lengths in the P(r) functions [11].

## NMR spectroscopy

NMR experiments are performed at 310 K on a Bruker Ultrashield 800 MHz and 500 MHz with a z-gradient cryoprobe. Protein samples are concentrated to 500 μM with 10 mM Tris (pH 7.5) and 7% D2O is added. All NMR spectra are processed with software package NMRPipe [12] and analyzed using NMRView [13]. Quadrature detection in the indirect dimensions is achieved either with States-TPPI or Echo/antiecho methods. Sequential backbone assignments are obtained from the following two- and three-dimensional experiments: 2D 1H-15N HSQC, 3D HNCO, HNCA, HN(CO)CACB, and HNCACB [14].

## Analytical ultracentrifugation

Sedimentation-velocity experiments were performed at 20˚C in a Beckman XL-I analytical ultracentrifuge using an An50Ti rotor. Aliquots of protein and reference buffer were loaded into a sedimentation-velocity cell equipped with a dual-sector charcoal-Epon centerpiece. The reference buffer used for studies of the ligand-free protein was an eluted fraction lacking protein, as determined by A280, obtained from size-exclusion chromatography of the protein sample. The reference buffer for centrifugation of CfPutA in the presence of ligands was the buffer from dialysis. Following a 2-h temperature equilibration, the sample was centrifuged at 35,000 rpm. The radial distribution of the sample was monitored with Rayleigh interference optics. Data were acquired at 2-min intervals for 300 radial scans. The data set was analyzed globally to obtain the sedimentation coefficient (c(s)) and molecular mass (c(M)) distributions using Sedfit.

# Results

## Generation of GlnBP mutants with different domain linker length

We create N170C (Asn 170 was mutated to Cys) mutants and introduce extra cysteine at C terminal that will form disulfide as an artificial liner, which controls the magnitude of domain closure based on our structural analysis. The N170 and C terminal residues are located at the interface between two domains. Our hypothesis is that the linkage of these two residues will perturb the domain closure that will affect ligand binding affinity. We introduce domain linker with 0 to 5 alanines that show different magnitude of restriction on domain closure. The linker composed of disulfide bond can be reversibly broken by reducing reagents.

## Characterization of ligand free GlnBP

The structural and thermodynamics change of GlnBP upon ligand binding are investigated by ITC, CD, NMR, and SAXS. Protein is purified by ion exchange and gel filtration chromatography. The purity is checked by SDS-PAGE, which shows a single band at 25 KDa. Overexpressed GlnBP is a mixture of ligand-free and bound GlnBP, which could be inferred from the shoulder peak on ion exchange chromatography. The collections are pooled and treated with guanidine hydrochloride to disrupt the ligand binding pocket and release ligand. The sample is loaded into the desalting column while maintaining 2 M guanidine during the process of separation. Then, we exchange the HPLC buffer with 10 mM Tris and run the sample through desalting column again to remove guanidine (S1 Fig).

## NMR assignments and ligand binding characterized by solution NMR

NMR methods have been developed to measure structural changes of GlnBP [9]. However, the complete backbone assignment of ligand-free GlnBP is still missing and 92% backbone assignment is achieved in this study. Assignment starts from residues with unique chemical shifts like alanine (average 19.11 ppm for Cβ), serine and threonine (average 63.74 ppm and 69.62 ppm for Cβ respectively), and glycine (average 45.38 ppm for Cα) (S2 Fig). Connectivity between residues was established initially using MARS [15] in an iterative manner, adding or adjusting assignments manually. Structural changes of GlnBP upon ligand binding are shown by dramatic change in heteronuclear single quantum correlation spectroscopy (HSQC) spectra of ligand-free and bound GlnBP (BMRB ID: 10171). Residues that are far from ligand binding sites also show chemical shift perturbation, which means that the binding event not only change the structure of binding pocket but change the whole protein structure (Fig 2A–2B).

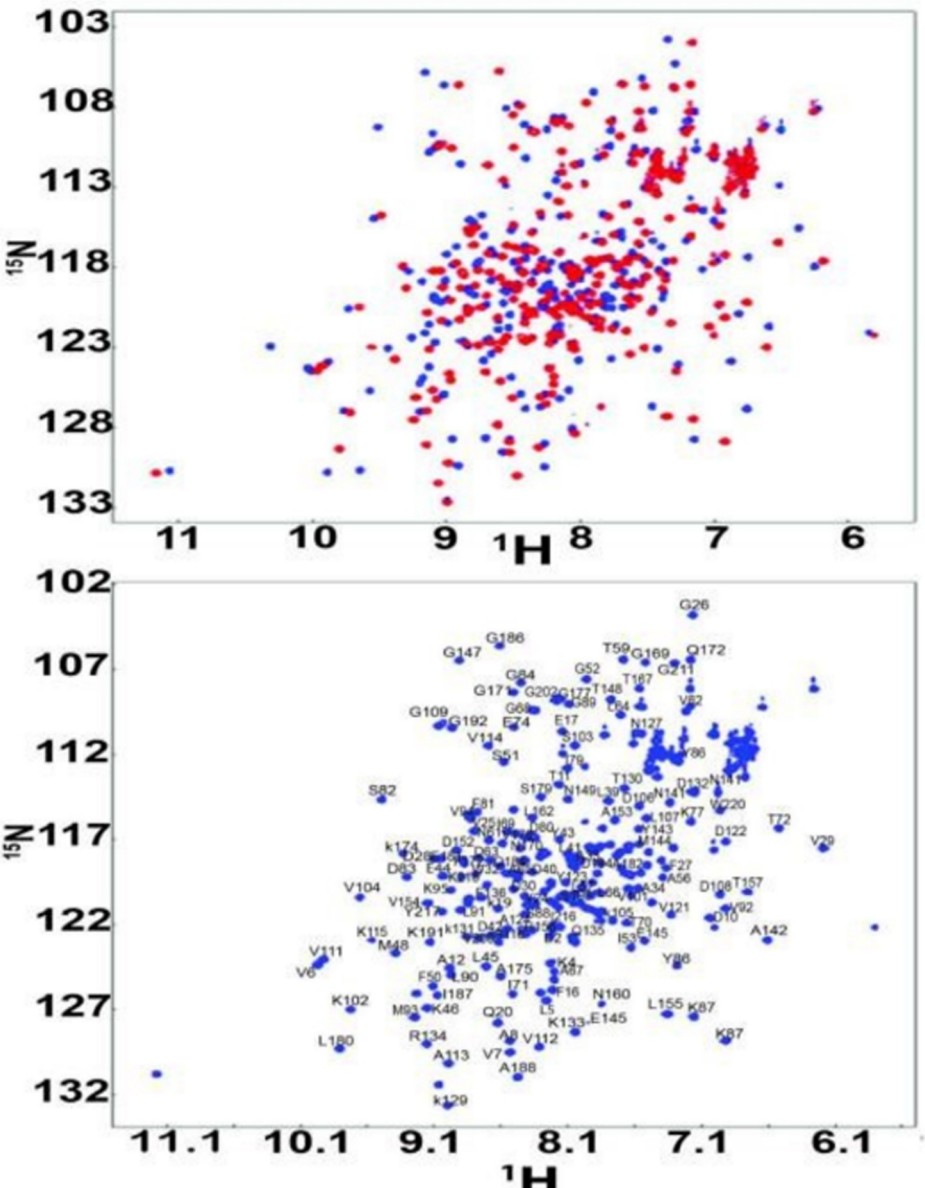

**Fig 2.** (a) Overlay of HSQC spectra for ligand-bound and free GlnBP. The red spectrum represents ligand-bound GlnBP and the blue one is GlnBP in ligand-free form; (b) HSQC of ligand-free GlnBP with the assignments.

## Gradual structural changes of GlnBP after ligand binding

Sedimentation velocity of analytical ultracentrifuge is used to check the aggregates of GlnBP at different concentrations. GlnBP is primarily monomeric, but inference data shows the presence of aggregates and the amount of aggregate appears to depend on the loading concentration from 0.1% at 83 μM to 2% at 1480 μM. SAXS is sensitive to tertiary and quarterly structural changes that is suitable for monitoring the conformational changes in ligand binding. The protein concentration for SAXS is 197 μM in 10 mM Tris, which corresponds to 4.9 mg/ml. Guinier analysis is used to determine the overall radius of gyration, Rg, and the concentration-normalized forward scattering intensity, I(0)/c, which are functions of spatial size

and molecular mass respectively. The Guinier plots are linear at the low q-range at all concentrations, indicating that GlnBP has a well-defined size under aqueous condition and radiation-induced aggregation is not an issue. Rg and I(0)/c, determined from linear fitting to the data over the q range satisfying the q $^*$ Rg < 1.3 condition (q = 0.0146 to 0.0416 Å-1), display very little concentration dependence, indicating that no change in oligomeric state occurs over the investigated concentration range (Fig 3A and 3B). Structural changes of GlnBP are also shown by SAXS ΔP(r) curves. With the increase in ligand concentration, the gradual changes are shown on ΔP(r) curves (Fig 3C and 3D). The gradual changes indicate the gradual occupation of binding pockets after increase in ligand concentration. The conformational changes are not abrupt and the magnitude of changes correlate with ligand binding.

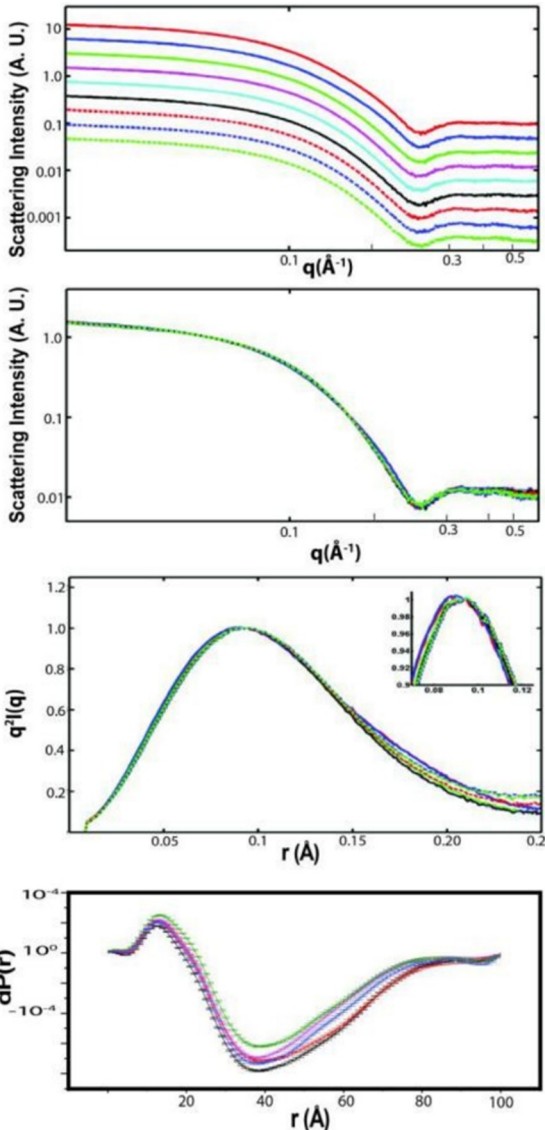

**Fig 3. Small angle X-ray scattering of GlnBP.** (a) Offset of I(q) vs q plot for GlnBP with different concentrations of ligand. Red line is 400 μM and green line is 0 μM. The increment is 50 μM from 0 to 400 μM; (b) Overlay of I(q) vs q plots for GlnBP with different concentrations of ligand; (c) Pair distance distribution function P(r) plot of GlnBP with different concentrations of ligands; (d) ΔP(r) shows the difference of conformational changes when different concentrations of ligand were titrated into the GlnBP protein solution.

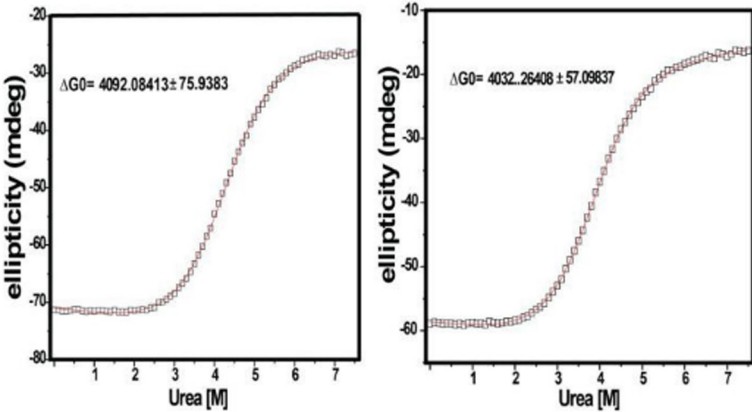

**Fig 4. CD Urea titration.** (a) 10 M urea was titrated into 5 M ligand bound and (b) ligand free GlnBP solution.

### Thermal stability of GlnBP in ligand free and bound state

Thermo stability is measured with urea CD titration (Fig 4A–4B). The difference of Gibbs free energy between ligand-free and bound GlnBP is 59.82 kJ/mol and the energy of hydrogen bond is 5–30 kJ/mol, which indicates that ligand binding introduces extra hydrogen bonds and van der Walls interactions that account for the energy difference. Tertiary structure changes dramatically but secondary structures are maintained, which explain the higher energy level of ligand-free GlnBP. GlnBP is rigid with [10] 15N NOE values around 0.9 except the N and C terminals and some residues that show slightly higher flexibility (data not shown).

### Domain closure affects GlnBP ligand binding affinity

The ligand binding affinity is measured with ITC for wild-type and mutants of GlnBP. Mutants with different ligand binding affinity are created by introducing a polypeptide linker composed of different numbers of alanine residue from 0 to 5 residues. The smaller side chain of alanine reduces the possibility of creating steric hindrance. The length of the linker between N170C (Asn 170 was mutated to Cys) and C terminal cysteine controls the magnitude of domain closure based on our structural analysis. The formation of disulfide bond is verified by Ellman's assay and mass spectrometry. The formation of disulfide bond decreases molecular weight by 2 due to the formation of new polypeptide bond, which can be verified by the molecular weight of GlnBP mutant with four alanine liner. Reduction of disulfide bond changes the molecular weight from 25043 to 25041 (S3 Fig). The correlation between domain closure and ligand binding is observed since shorter linker mutants have a lower binding affinity, which indicates that ligand binding triggers the protein conformational change and induces domain closure to create a perfect binding site for ligand. The mutants with artificial linker can be reversed by reducing reagent. The presence of linker inhibits domain closure and reduces ligand binding affinity. Breakage of linker restore the ligand binding comparable to wild type protein (Fig 5, S4 Fig).

## Discussion

GlnBP is a soluble glutamine binding protein swimming in the periplasmic space and escorts free glutamine into membrane transporter. Ligand binding to GlnBP triggers a large-scale conformational change and two domains close to pack the glutamine into the ligand binding pocket. The ligand binding affinity determines the efficiency of transportation in vivo and the

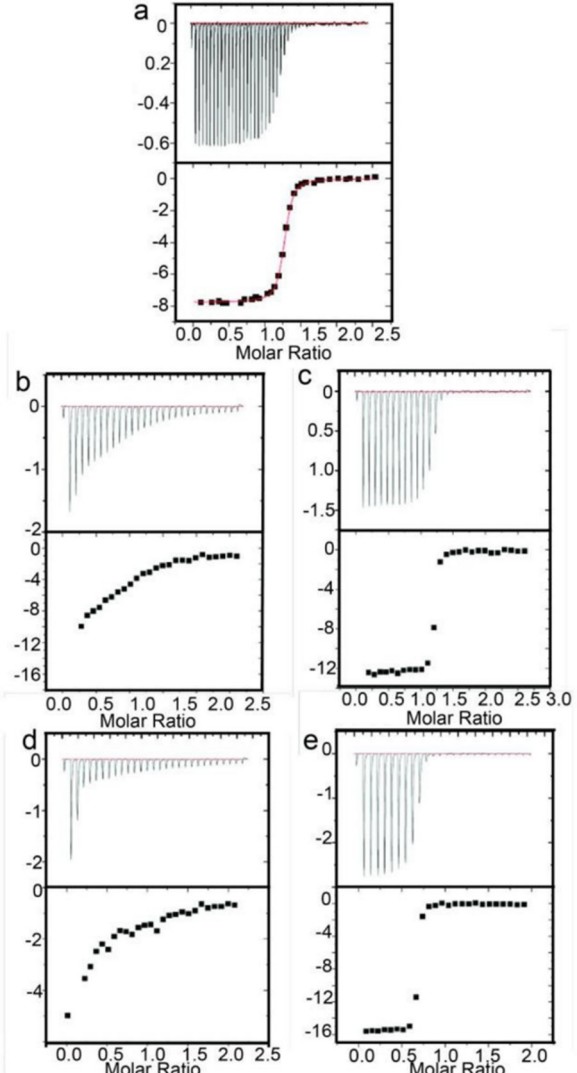

**Fig 5. ITC of ligand binding.** (a) ITC for wild-type GlnBP; (b) ITC for GlnBP 2A mutant and (c) with the addition of reducing reagents (DTT); (d) ITC for GlnBP 3A mutant and (e) with the addition of reducing reagents (DTT).

application of GlnBP in the field of biosensors. Dynamics of GlnBP have been characterized by NMR [5] and molecular dynamics simulation [16]. However, the correlation of conformation change and ligand binding affinity remains elusive. We use NMR, SAXS, and ITC to characterize the ligand binding of GlnBP and corresponding structural changes. The complementary nature of these techniques provides biophysical insights of molecular binding. We generate GlnBP mutants with different domain linker length to investigate the perturbation of domain closure capability on protein ligand binding. We find that mutants of GlnBP with different domain closure capability show different ligand binding property. Longer linker allows the free domain closure that has the highest ligand binding affinity. Shorter domain linker blocks domain closure that has the lowest binding affinity. Breakage of disulfide bond release the restriction on domain closure and restore ligand binding affinity. We discover a reversible way to control domain closure and ligand binding. The study shows the correlation between ligand binding affinity and domain closure, which is the major conformational change upon ligand binding and determinant for ligand binding.

The structure of GlnBP is strikingly similar to the ligand binding core (S1S2) of ionotropic glutamate receptor. Ionotropic glutamate receptors (iGluR) are major excitatory receptors in the vertebrate central nervous system. The ligand-binding core of iGluR is formed by polypeptide fragments S1 and S2, which could be coupled with a 13-residue liner to form a new construct S1S2. S1S2 has the same binding property as wild-type protein. The homotetrameric structure of intact iGluR was resolved recently and it demonstrates that S1S2 construct has identical structure as the ligand-binding core of intact receptor. Structural and dynamic information obtained from GlnBP could shed light on the function of iGluR and provides hints about the structural conservation even if their sequences have diverged dramatically. Potential drug development can target the domain closure to perturb the ligand binding, which provide theoretical foundation for targeted therapeutics.

## Supporting information

**S1 Fig. Purification of GlnBP and removal of Gln ligand through desalting column.**
(TIF)

**S2 Fig. Representative 3D NMR for GlnBP backbone chemical shift assignment.**
(TIF)

**S3 Fig. Mass spectrometer characterization of GlnBP in intact and reduce format.** The corresponding molecular weight is 25041 and 25043.
(TIF)

**S4 Fig. ITC of GlnBP with 4 and 5 alanine linkers.**
(TIF)

## Acknowledgments

The use of the Advanced Photon Source, an Office of Science User Facility operated for the U. S. Department of Energy (DOE) Office of Science by Argonne National Laboratory, was supported by the U.S. DOE.

## Author Contributions

**Conceptualization:** Peng Jia.

**Data curation:** Xiaobing Zuo.

**Formal analysis:** Qun Chen.

**Funding acquisition:** Peiwu Qin.

**Investigation:** Chengming Yang.

**Supervision:** Jin Chen, Peiwu Qin, Dongmei Yu.

**Writing – original draft:** Fang Li, Chuhui Wang, Danyu Yang.

**Writing – review & editing:** Fang Li, Peiwu Qin, Jin Xu, Baogang Xing, Ying Liu, Linling Li, Dongmei Yu.

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
