## [Decision Letter · Decision Letter 0]

29 Jan 2021

PONE-D-20-37100

Combination of SAXS and NMR to monitor the conformation changes of E. Coli glutaminine binding protein upon ligand binding

PLOS ONE

Dear Dr. Yu,

Thank you for submitting your manuscript to PLOS ONE. Let me apologize for the slowness of the review process. The holiday season and COVID19 has affected most schedules.  After careful consideration, we feel that it has merit but does not fully meet PLOS ONE’s publication criteria as it currently stands. As you can see, one reviewer had fair but strong objections to the manuscript.  Despite their recommendation, I will recommend major revision because if these objections can be properly addressed, it would make the manuscript stronger. Also the other reviewer was less critical but you can see if their review that some of their concerns also resonant with comments of the other reviewer. Therefore, we invite you to submit a revised version of the manuscript that addresses the points raised during the review process.

We look forward to receiving your revised manuscript.

Kind regards,

Michael Massiah

Academic Editor

PLOS ONE

Journal Requirements:

"This work was funded by National Natural Science Foundation of China (31970742, 81972854, 81802626), Science,

Technology and Innovation Commission of Shenzhen Municipality (JSGG20191129110812), and Southern

University of Science and Technology Hospital Dean Research Fund (2020-A4)."

"Prof. Peiwu Qin received National Natural Science Foundation of China (31970742, 81972854, 81802626, http://www.nsfc.gov.cn/english/site_1/index.html), and Science, Technology and Innovation Commission of Shenzhen Municipality (JSGG20191129110812, http://stic.sz.gov.cn/). The funders had no role in study design, data collection and analysis, decision to publish, or preparation of the manuscript."

Reviewers' comments:

Reviewer's Responses to Questions

**Comments to the Author**

1. Is the manuscript technically sound, and do the data support the conclusions?

Reviewer #1: Yes

Reviewer #2: No

2. Has the statistical analysis been performed appropriately and rigorously? 

Reviewer #1: Yes

Reviewer #2: No

3. Have the authors made all data underlying the findings in their manuscript fully available?

Reviewer #1: No

Reviewer #2: No

4. Is the manuscript presented in an intelligible fashion and written in standard English?

Reviewer #1: Yes

Reviewer #2: No

5. Review Comments to the Author

Reviewer #1: The authors have investigated the configurational modifications of the glutamine binding protein (GlnBP) upon ligand binding by NMR and SAXS in order to determine the correlation between ligand binding affinity and domain closure. Dynamics of the hinge region, located between the two domains that constitute the protein, induce open-to-close conformation upon ligand binding and can be determined by variation of ligand concentration and mutants with linkers of different lengths. Although a complete structure determination of the conformational variations has not been achieved, the authors present a relevant structural characterization that contributes to the knowledge in the corresponding research field.

This article is of interest, well-written, and the experiments seem to be well-conducted, with few issues that need to be addressed that are listed below.

- The authors could use SAXS analysis to obtain envelope models for the different domain closure configurations and complement NMR data. Different methodologies for protein structure determination by NMR and SAXS conjugation can be implemented, as described at Rodriguez-Zamora, P. "Conjugation of NMR and SAXS for flexible and multidomain protein structure determination: from sample preparation to model refinement." Progress in biophysics and molecular biology 150 (2020): 140-144. If such data is available it should be included in the article, otherwise it would be recommended to mention the possibility of an NMR-SAXS conjugated analysis as a perspective for this work.

- A figure showing the protein hinge section and the linker introduction is required to illustrate the region of interest.

- Figure 5 is missing a color code that clarifies the results obtained by SAXS.

- Few grammar errors need to be corrected throughout the manuscript. A PDF copy with some of those errors highlighted is attached.

Reviewer #2: The manuscript by Chen et al. described the structure changes of addition mutant of E. coli. Glutamine binding protein (GlnBP) upon ligand (glutamine) binding. The experimental methods of ITC, NMR, SAXS etc. were applied. The conclusion was the change of binding affinity was associated with quaternary structure change (inter-domain) caused by the lengthened linker mutant. While the inter-domain reorganization may contribute to the binding affinity changes, the contribution is one of many factors that could affect binding affinity. Potentially, electrostatic and hydrophobic interactions between protein side chains and Gln should dominate the interaction, not discussed in the manuscript. The authors don’t seem to have a clear experimental design or conclusion. In general, for a study like this, the hypotheses should be raised at the beginning, followed by experimental demonstration, result interpretation and final conclusion. Now, the manuscript lacks coherent reasoning or conclusion. In the end, there is no take home message from the study. The manuscript is highly immature and should not be published in the current format. Major issues are listed below.

1. The result did not provide any quantitative correlation between ITC results and any others like SAXS or NMR. Therefore, the science advance on this binding topic is minimal.

2. The scattering of “Results” part was not acceptable, where sections were missing. The authors appeared to be random in narratively describing the results. Normally, sections of mutation selection, structure changes, correlation between xxx and yyy etc, should be in “Results”. For a biophysics study like this, a table of GlnBP protein of WT, mutant 1, mutant 2 … should be presented. This should be followed by structural characterization (NMR., SAXS, SV-AUC) at free and binding states of each protein. The authors should then infer from analytical results on the structural change of GlnBP at secondary, tertiary, quaternary structure levels. For example, NMR chemical shift could be sensitive to secondary and tertiary structure, relaxation could be sensitive to quaternary and oligomerization, SAXS could be sensitive to quaternary or domain structure, SV-AUC is for oligomerization. All of these standard reasonings are missing in the manuscript.

3. The 1st paragraph in “Results” belongs to method description.

4. Page 2 Paragraph 1, the statement “two domain linked by a flexible “hinge” region” was not correct. The 2 domains were linked by 2 rigid beta-strands per crystal structure.

5. The method section missed the description of SV-AUC method.

6. PLOS authors have the option to publish the peer review history of their article (what does this mean?). If published, this will include your full peer review and any attached files.

Reviewer #1: No

Reviewer #2: No

---

## [Author Response · Author response to Decision Letter 0]

24 Aug 2021

Reviewer #1: The authors have investigated the configurational modifications of the glutamine binding protein (GlnBP) upon ligand binding by NMR and SAXS in order to determine the correlation between ligand binding affinity and domain closure. Dynamics of the hinge region, located between the two domains that constitute the protein, induce open-to-close conformation upon ligand binding and can be determined by variation of ligand concentration and mutants with linkers of different lengths. Although a complete structure determination of the conformational variations has not been achieved, the authors present a relevant structural characterization that contributes to the knowledge in the corresponding research field.

This article is of interest, well-written, and the experiments seem to be well-conducted, with few issues that need to be addressed that are listed below.

1- The authors could use SAXS analysis to obtain envelope models for the different domain closure configurations and complement NMR data. Different methodologies for protein structure determination by NMR and SAXS conjugation can be implemented, as described at Rodriguez-Zamora, P. "Conjugation of NMR and SAXS for flexible and multidomain protein structure determination: from sample preparation to model refinement." Progress in biophysics and molecular biology 150 (2020): 140-144. If such data is available it should be included in the article, otherwise it would be recommended to mention the possibility of an NMR-SAXS conjugated analysis as a perspective for this work.

We thank reviewer for the valuable suggestions. We conducted SAXS experiments at Argonne National Laboratory and the data was analyzed with the assistance of the staff scientist at Argonne. Both the ligand bound and free GlnBP have been solved by X-ray crystallography. Thus, we feel it is not necessary to get the shape of molecules since atomic structure is available. SAXS is capable of monitoring the gradual structural changes upon ligand binding, which is not possible by x-ray diffraction. The structural changes of GlnBP were confirmed and monitored from two complementary techniques, which is enough for our purpose to claim the gradual domain closure upon ligand binding. 

2- A figure showing the protein hinge section and the linker introduction is required to illustrate the region of interest.

For the GlnBP structure, we add figure 1c to illustrate the linker region of interest. The green indicates the large domain of GlnBP, the magenta represents the small domain, and the hinges we are interested (residues 85–89 and residues 181–185) is highlighted in red. We introduce the linker between position N170 and C terminal labeled with cyan color.

 Red line is 400 μM and green line is 0 μM. The increment is 50 μM from 0 to 400 μM.

- Few grammar errors need to be corrected throughout the manuscript. A PDF copy with some of those errors highlighted is attached.

We thank reviewer for pinpointing the grammar errors and we correct them.

Reviewer #2: The manuscript by Chen et al. described the structure changes of addition mutant of E. coli. Glutamine binding protein (GlnBP) upon ligand (glutamine) binding. The experimental methods of ITC, NMR, SAXS etc. were applied. The conclusion was the change of binding affinity was associated with quaternary structure change (inter-domain) caused by the lengthened linker mutant. While the inter-domain reorganization may contribute to the binding affinity changes, the contribution is one of many factors that could affect binding affinity. Potentially, electrostatic and hydrophobic interactions between protein side chains and Gln should dominate the interaction, not discussed in the manuscript. The authors don’t seem to have a clear experimental design or conclusion. In general, for a study like this, the hypotheses should be raised at the beginning, followed by experimental demonstration, result interpretation and final conclusion. Now, the manuscript lacks coherent reasoning or conclusion. In the end, there is no take home message from the study. The manuscript is highly immature and should not be published in the current format. Major issues are listed below.

We thank reviewer for the valuable comments. We did make a hypothesis that domain closure is critical for ligand binding at the beginning. In order to approve this hypothesis, we create a series of artificial linker to change the domain closure. Then, we study how the ligand binding is affected by these mutants. The reviewer is absolutely correct that electrostatic and hydrophobic interactions between side chain and Gln influence ligand binding. However, this has been demonstrated in many ligand binding proteins. Domain closure or linker affect ligand binding, which has never been investigated to our knowledge. We maintain the protein sequence to exclude the possibility that electrostatic or hydrophobic interaction determines ligand binding. 

We add a few sentences in the discussion to highlight the significance of this study. We use NMR, SAXS, and ITC to characterize the ligand binding of GlnBP and corresponding structural changes. The complementary nature of these techniques provide biophysical insights of molecular binding. We generate GlnBP mutants with different domain linker length to investigate the perturbation of domain closure capability on protein ligand binding. We find that mutants of GlnBP with different domain closure capability show different ligand binding property. Longer linker allows the free domain closure that has the highest ligand binding affinity. Shorter domain linker blocks domain closure that has the lowest binding affinity. Breakage of disulfide bond releases the restriction on domain closure and restore ligand binding affinity. We discover a reversible way to control domain closure and ligand binding. The study shows the correlation between ligand binding affinity and domain closure for the first time, which is the major conformational change upon ligand binding and determinant for ligand binding. 

1. The result did not provide any quantitative correlation between ITC results and any others like SAXS or NMR. Therefore, the science advance on this binding topic is minimal.

ITC, NMR, and SAXS study different properties of protein complex. ITC studies the ligand binding affinity that shows different linker length perturbs the ligand binding affinity. Then, we use SAXS to show that ligand binding gradually trigger domain closure and protein conformation changes. The results from three techniques are complementary and complete to characterize the ligand binding. SAXS and NMR both provide aqueous structural information and we have demonstrated the ligand binding induced conformation changes from these two techniques. Since crystal structures of GlnBP in ligand free and bound state are available, we skip the generation of protein shape by SAXS and NMR structure determination. 

2. The scattering of “Results” part was not acceptable, where sections were missing. The authors appeared to be random in narratively describing the results. Normally, sections of mutation selection, structure changes, correlation between xxx and yyy etc, should be in “Results”. For a biophysics study like this, a table of GlnBP protein of WT, mutant 1, mutant 2 … should be presented. This should be followed by structural characterization (NMR., SAXS, SV-AUC) at free and binding states of each protein. The authors should then infer from analytical results on the structural change of GlnBP at secondary, tertiary, quaternary structure levels. For example, NMR chemical shift could be sensitive to secondary and tertiary structure, relaxation could be sensitive to quaternary and oligomerization, SAXS could be sensitive to quaternary or domain structure, SV-AUC is for oligomerization. All of these standard reasonings are missing in the manuscript.

We thank reviewer for this valuable comment. We add section titles in the results part. We add one more section to introduce the design logic behind the mutants with different length of domain linker as the following:

Generation of GlnBP mutants with different domain linker length

We create N170C (Asn 170 was mutated to Cys) mutant and introduce extra cysteine at C terminal that will form disulfide as an artificial liner, which controls the magnitude of domain closure based on our structural analysis. The N170 and C terminal residues are located at the interface between two domains. Our hypothesis is that the linkage of these two residues will perturb the domain closure that will affect ligand binding affinity. We introduce domain linker with 0 to 5 alanine that show different magnitude of restriction on domain closure. The linker composed of disulfide bond can be reversibly broken by reducing reagents. 

We add a few sentences to each section of results to give detail explanation about the physical and biological meaning of the data.

We agree with the reviewer that correlation between xxx and yyy etc, should be in “Results”. We show the results in a reversible assay. The mutants we generate can introduce a reversible linker. The ligand binding affinity is perturbed and restored after domain linker formation or breakage. Reversibility is a stronger evidence than correlation analysis. The biophysical measurements of diverse GlnBP mutants are not complete between the structural changes are minimal and ligand binding is the most important property we are interested. 

3. The 1st paragraph in “Results” belongs to method description.

The first paragraph illustrates how we prepare and characterize ligand free GlnBP since GlnBP from cell lysate contain both ligand free and bound GlnBP. We treat the purified sample with detergent and remove the ligand. If the sample is a mixture of ligand free and bound format, the following binding assay will have great measurement errors. 

4. Page 2 Paragraph 1, the statement “two domain linked by a flexible “hinge” region” was not correct.

We changed to “The 2 domains were linked by 2 rigid β-strands per crystal structure.” 

5. The method section missed the description of SV-AUC method.

We thank reviewer for pointing this out. We add one paragraph to describe the SV-AUC method as the following:

Analytical ultracentrifugation

Sedimentation-velocity experiments were performed at 20 °C in a Beckman XL-I analytical ultracentrifuge using an An50Ti rotor. Aliquots of protein and reference buffer were loaded into a sedimentation-velocity cell equipped with a dual-sector charcoal-Epon centerpiece. The reference buffer used for studies of the ligand-free protein was an eluted fraction lacking protein, as determined by A280, obtained from size-exclusion chromatography of the protein sample. The reference buffer for centrifugation of CfPutA in the presence of ligands was the buffer from dialysis. Following a 2-h temperature equilibration, the sample was centrifuged at 35,000 rpm. The radial distribution of the sample was monitored with Rayleigh interference optics. Data were acquired at 2-min intervals for 300 radial scans. The data set was analyzed globally to obtain the sedimentation coefficient (c(s)) and molecular mass (c(M)) distributions using Sedfit.

---

## [Decision Letter · Decision Letter 1]

16 Sep 2021

PONE-D-20-37100R1Combination of SAXS and NMR to monitor the conformation changes of E. Coli glutaminine binding protein upon ligand bindingPLOS ONE

Dear Dr. Yu,

Thank you for submitting your manuscript to PLOS ONE. As you can see, one of the reviewers still has serious concerns, some of which appears to not be address in this revision. Can you please address of these concerns. Thanks. After careful consideration, we feel that it has merit but does not fully meet PLOS ONE’s publication criteria as it currently stands. Therefore, we invite you to submit a revised version of the manuscript that addresses the points raised during the review process.

We look forward to receiving your revised manuscript.

Kind regards,

Michael Massiah

Academic Editor

PLOS ONE

Journal Requirements:

Reviewers' comments:

Reviewer's Responses to Questions

**Comments to the Author**

1. If the authors have adequately addressed your comments raised in a previous round of review and you feel that this manuscript is now acceptable for publication, you may indicate that here to bypass the “Comments to the Author” section, enter your conflict of interest statement in the “Confidential to Editor” section, and submit your "Accept" recommendation.

Reviewer #1: All comments have been addressed

Reviewer #2: (No Response)

2. Is the manuscript technically sound, and do the data support the conclusions?

Reviewer #1: Partly

Reviewer #2: No

3. Has the statistical analysis been performed appropriately and rigorously? 

Reviewer #1: N/A

Reviewer #2: No

4. Have the authors made all data underlying the findings in their manuscript fully available?

Reviewer #1: Yes

Reviewer #2: No

5. Is the manuscript presented in an intelligible fashion and written in standard English?

Reviewer #1: Yes

Reviewer #2: No

6. Review Comments to the Author

Reviewer #1: (No Response)

Reviewer #2: The manuscript by Chen et al. remained immature after the revision. Though a series of artificial linker mutants were generated, the mutant protein was only subject to ITC study, not NMR, SAXS or AUC. The NMR study on the same WT protein and ligand has been extensively studied by others before. The authors’ current studies were isolated on different proteins, and lacked quantitative results or correlation. This type of scattering in study was not reflected in “title”, which implied NMR and SAXS were applied on mutants, actually not. A table listing protein variants, methods and results were suggested to the authors, but authors did not take the suggestion. The main conclusion of longer linker introducing tighter binding between protein and ligand was commonly assumed in protein biochemistry because of less steric restriction. The study using ITC only verifies the common knowledge. A scientific approach should drive from the avenue of free energy landscape to interpret results and give more quantitative insight to molecular interaction. Due to the less quantitation and correlation, the manuscript did not deliver any robust new findings that merit the publication.

7. PLOS authors have the option to publish the peer review history of their article (what does this mean?). If published, this will include your full peer review and any attached files.

Reviewer #1: No

Reviewer #2: No

---

## [Author Response · Author response to Decision Letter 1]

13 Nov 2021

We addressed the first reviewer’s comments, and the following is our response to the second reviewer’s concerns. We really thank reviewer for above points, in our abstract, we want to approve hypothesis that the freedom of closure is vital for ligand binding, and our experiments approved this hypothesis. We change the manuscript title to “Reversible domain closure modulates GlnBP ligand binding affinity” that summarize the chief finding more appropriately. The mutants are variants of GlnBP with different linker length from 05 alanine residues. Thus, we think it is not necessary to list these simple mutants with one more table. The reviewer comments that “The study using ITC only verifies the common knowledge”. Our manuscript is the first study to investigate the correlation between domain closure and ligand binding. ITC for ligand binding is a common knowledge. The usage of ITC in characterization of domain closure is novel. We thank review for the suggestion about evergy landscape. Our focus is the relationship between domain closure and ligand binding affinity. Gibbs energy changes is a concrete demonstration or explaination of our conclusion. ITC and SAXS are direct measurements to approve the relationship between domain opening and ligand binding.

---

## [Editor Report · Decision Letter 2]

13 Jan 2022

Reversible domain closure modulates GlnBP ligand binding affinity

PONE-D-20-37100R2

Dear Dr. Yu,

We’re pleased to inform you that your manuscript has been judged scientifically suitable for publication and will be formally accepted for publication once it meets all outstanding technical requirements.

Kind regards,

Michael Massiah

Academic Editor

PLOS ONE
---

## [Editor Report · Acceptance letter]

12 Apr 2022

PONE-D-20-37100R2 

Reversible domain closure modulates GlnBP ligand binding affinity 

Dear Dr. Yu:

I'm pleased to inform you that your manuscript has been deemed suitable for publication in PLOS ONE. Congratulations! Your manuscript is now with our production department. 

Kind regards, 

on behalf of

Dr. Michael Massiah 

Academic Editor

PLOS ONE